# Synthesis of Graphene Quantum Dots Coupled to Au Nanoparticles: A Facile and Versatile Route Using Different Carbon Sources

David Ibarra [1,†,‡] , Oxana Kharissova [2,‡] and Idalia Gomez [1,*,‡]

1  Facultad de Ciencias Químicas, Universidad Autónoma de Nuevo León, Mexico CP 66455, Mexico
2  Facultad de Ciencias de Físico-Matemáticas, Universidad Aútonoma de Nuevo León,
   Mexico CP 66451, Mexico
*  Correspondence: maria.gomezd@uanl.edu.mx; Tel.: +52-8183294000 (ext. 6240)
†  Current address: Laboratorio de Materiales I, Facultad de Ciencias Químicas, Universidad Autónoma de
   Nuevo León, Cd. Universitaria, México CP 66455, Mexico.
‡  These authors contributed equally to this work.

**Abstract:** A top-down synthesis of graphene quantum dots (GQD) was carried out by hydrothermal method from different carbon sources (graphene, multi-walled carbon nanotubes, and black carbon) and $H_2O_2$ as an oxidizing agent, with an N source added in the reaction to modify the chemical surface of the GQD, giving rise to the nanomaterial N-GQD. The modified chemical surface of N-GQD partially allowed the nucleation and coupling of gold nanoparticles from a $HAuCl_4$ solution. The X-ray diffraction spectrogram confirms the amorphization of the precursor materials, while the functionalized surface of N-GQD was characterized through UV-Vis, Fourier transform infrared, and photoluminescense spectrometry; TEM and FE-SEM show particle sizes between 8 and 15 nm. N-GQD@AuNP presence can be confirmed by UV-Vis spectroscopy and TEM analysis, showing partial coupling and nanoparticle nucleation of Au in the structure with particle sizes between 20 and 40 nm.

**Keywords:** hydrothermal; quantum dots; nanosensors; carbon nanoparticles; top-down





## 1. Introduction

Carbon-based nanomaterials have experienced great development in research since the discovery of graphene due to the versatility of their properties, which are highly dependent on their atomic structure, composition, and interaction with other materials [1–3]. Graphene quantum dots (GQD) are an example of these graphene-based nanomaterials; the space of their structure is confined to nanometer scales (around 10 nm) in its three dimensions, thus having a single atomic layer thickness [1,4]. Though it is a relatively recent material, its quantum effects are theoretically well known in relation to its electronic properties (which are mainly edge effects that introduce low energy states), and various methods have been developed for their synthesis; it is essential to facilitate their obtaining in order to find useful applications for this novel nanomaterial [5,6].

GQD exhibit properties such as low toxicity, stability, excellent electrical conductivity, large specific surface area, low cost synthesis, green synthesis, and biocompatibility; furthermore GQD exhibit fluorescence because of quantum confinement, which makes them excellent candidates as optoelectronic sensors, biosensors, or electrochemical biosensors on their own [7]. When the chemical surface of GQD is modified, itsproperties change too [8]. Functional groups on the surface of GQD, lateral size, dopants, and localized domains can alter the electronic energy transitions and cause changes in fluorescence peak wavelength, peak width, and peak position [5,9]. For instance, doping GQD with N improves their fluorescence properties, making them more stable and providing more active sites. In addition to changing the properties, the modification of the chemical surface of

the GQD converts the nanomaterial to a base for the coupling of nanoparticles, such as Au nanoparticles; the functional groups function as reducing agents and nucleation sites for metal nanoparticles [10–12]. AuNP have been very present in the areas of medicine, diagnostics, and pharmacology, being used as biomarkers; thus, their utilities are benefited by coupling to GQD, being soluble in water, being biologically compatible, improving electron donation and reception interactions, and preserving characteristic properties of AuNP such as surface plasmon resonance (which is their most useful feature) as well as their ease of synthesis [1,12–14].

The synthesis of GQD is mainly focused on two methodologies: bottom-up and top-down. Within the top-down, there is the oxidative cleavage that is carried out by means of a hydrothermal reaction. The reaction consists of exposing carbon materials to an oxidizing agent, such as hydrogen peroxide, at high temperatures. The radicals of the oxidizing agent react with the carbon materials, resulting in the formation of nanometer-sized graphene fragments [5,15]. This synthesis strategy is attractive due to its short reaction times (180 min at 200 °C) and easy preparation; in addition, it is considered a green synthesis because it uses precursors with low environmental impact such as black carbon and hydrogen peroxide (at hydrothermal conditions free the $H^-$ and $OH^-$ radicals and end up forming $H_2O$ as a residue), this top-down approach enables more straightforward purification of the final product [5,16]. Hydrogen peroxide has extensively been used in graphene research in order to introduce pores [17]. For hydrothermal reactions, $H_2O_2$ is an effective oxidizer that can cut the structure of graphene-based materials [16]. Although the use of top-down methodologies has been less common compared with bottom-up ones, since the work reported by Lu et al. (2017) [16] in 2017, due to difficulties in yield and morphology, they have managed to improve the results obtained as demonstrated by the work of Su et al. (2020) [18] in 2020, so it has come to be considered to be presented as a promising alternative synthesis pathway for this work.

In this work, the synthesis of N-GQD is performed via a hydrothermal reaction using a carbon source (black carbon, graphen, and MWCNT-OH in order to observe the variability of the results in function of the precursor materials), EDA (etilendiamine) as a nitrogen source, and $H_2O_2$ as an oxidizer agent. Nitrogen radicals on the chemical surface of N-GQD provoke the nanomaterial to perform the role of a reducing agent and start the nucleation of the AuNP by a simple addition of thetraclorauric acid with vigorous staring. The synthesized N-GQD exhibits fluorescence when exposed to UV radiation, and the N-GQD@AuNP nanocomposite experiment a change in their coloration in the solution at finished reaction, indicating the AuNP presence.

## 2. Materials and Methods

### 2.1. Sample Preparation

Table 1 shows the quantities of carbon sources used and their respective labels to measure 30 mg of carbon source.

**Table 1.** Different sources of carbon and the quantities measured.

| Carbon Source | Quantity | Label |
|---|---|---|
| Black carbon | 0.0367 g | D1 |
| MWCNT-OH | 0.0328 g | D2 |
| Graphene | 0.0305 g | D3 |

The experiments have been labeled with a key to differentiate each of them. Subsequently, in a typical procedure as described in the work by Lu et al. (2017) [16] and Su et al. (2020) [18], 1.3 mL of 30% $H_2O_2$ (wt%) and 7.5 mL of 99.99% (purity) EDA, both purchased from Sigma-Aldrich, were added to each of the experiments, to respect the reason of approximate proportions in weight of 1:13:225 (large amounts of EDA ensure the modification of the chemical surface of the products). EDA, as a reducing agent, has

a high affinity for carbon surfaces and can form strong covalent bonds with the carbon atoms, leading to the surface functionalization of GQD (EDA is in excess to ensure the modification of the chemical surface) [11,18]. In addition, EDA has a low molecular weight and is a liquid at room temperature, making it easy to handle and mix with other reactants. The solution is taken to the Teflon-lined hydrothermal reactor and completed to 30 mL of solution with distilled water. The experimental conditions were 180 °C for 6 h, extending the time in order to ensure the reaction (optimal conditions reported: 200°, 90 min.) [16], in a Thermo-Scientific furnace model Lindberg/Blue M - Moldatherm. To eliminate the excess of ammonia that could be formed, the rotary evaporator (Vertical IKA RV 10 Digital-V) was used with moderate heating (less than the boiling of water) enough to eliminate the mentioned solvent for around 10 min at 80 revolutions per minute. An ultrasonic bath (Kasalab model SKU: PS-40AL) was carried out to ensure that the solution was homogenized for 1 h. The N-GQD solution is ready to be characterized. During the experimental process, one of the replications of Sample D2 was accidentally exposed for 24 h to the hydrothermal treatment. It is considered that this could provide additional information of value to this project on how it affects the reaction time for the precursor MWCNT-OH. This sample is labeled as D2/24 h.

The coupling of AuNP in the structure was carried out under the optimal conditions mentioned in the work by Hai et al. (2018) [12].In a beaker, ice was placed to later mount a 25 mL ball flask with a magnetic stirrer held with a three-finger clamp. In the ball flask, 1 mL of the N-GQD solution was poured along with 3 mL of deionized water. It was left under stirring for 10 min in a hot plate magnetic stirring (ThermoFisher model Cumarec +). Subsequently, the previously prepared solution of $HAuCl_4$ $5 \times 10^{-3}$ mM was added dropwise until a change in color was observed. In this case, it went from a pale yellow to a bright one. The solution proceeded to be analyzed.

### 2.2. Characterization

The precursors of N-GQD and the synthesized N-GQD were analyzed by X-ray diffraction (XRD) Bruker, model D2 PHASER. The produced N-GQD samples were observed by recording pictures of the aqueous solution in the dark under ultraviolet (UV) light with a wavelength of 365 nm produced by a commercial UV LED flashlight (Steren, model No. 560). The UV-Vis optical absorption spectra were measured by a Shimadzu model UV-1800 spectrometer. Fluorescence spectra were recorded in a Perkin Elmer LS 55 Fluorescence spectrometer. FT-IR was used to characterize the functional groups in a Perkin Elmer SPEC-TRUM TWO spectrometer. The microscopic analysis was carried out using an Angstrom Advanced-AA 3000 FE-SEM and a Hitachi H-9500 High Resolution TEM.

### 3. Results

#### 3.1. Chemical and Structural Characterization of N-GQD

The changes in the precursor matter can first be observed in XRD spectrometry (Figure 1), where if we compare the spectra of the precursors against the spectra of the experiments after hydrothermal treatment, we can observe (in general) a decrease in intensity and broadening of the more pronounced peaks.

It can be seen in the series of graphs in Figure 1 where the precursor materials have a very pronounced peak in the $2\theta$ range of 25–27°, which after treatment is reduced, indicating the formation of nanocrystals and that the dispersion of their domains causes broadening of the signal. Although carbon black is the least crystalline material, we can still observe this phenomenon clearly. In the literature on instrumental analysis and nanotechnology, we can find this amorphization process described, which in this case is related to obtaining a nanomaterial [19,20].

It is worth noting that the process of amorphization does not necessarily result in a reduction and broadening of the signal specifically at the main peaks of the precursor material. Instead, there may be a pattern in which the main peaks shift within a specific angle range, such as 25–30°, after hydrothermal treatment. This pattern has been observed



in other precursor materials as well as in the graphene sample in Figure 1c, which has a diffraction peak at around 30° instead of 26° after treatment. This shift in peak position may also indicate the formation of nanocrystals and the dispersity of their domains, which contributes to the broadening of the overall signal.

Regarding the XRD spectrum of the carbon black precursor shown in Figure 1a, the peaks at 2θ angles of 10 and 20 degrees may be related to the presence of organic compounds or impurities resulting from the combustion of burned tires, from which the carbon black sample was obtained. This hypothesis should be further investigated.

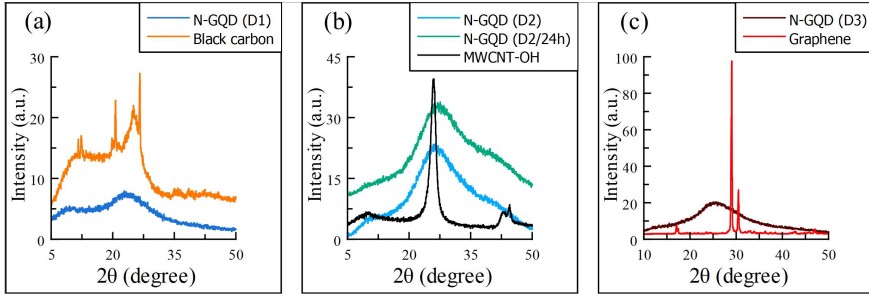

**Figure 1.** X-ray diffractograms of GQD synthesized using (**a**) black carbon, (**b**) MWCNT-OH, and (**c**) graphene carbon sources for pre and post hydrothermal treatmeant, showing a clearly amorphization of the precursor materials (the y-axis was modified to show better this effect). In addition, it is observed a very similar diffractogram (**b**) for the Sample D2 and its variation D2/24 h.

For FT-IR analysis in Figure 2, IR spectra shown for all four samples, to be quite similar. It can first observe a strong and broad signal at wavenumber 3400 cm$^{-1}$ indicating the presence of $-OH$ and $-NH_2$ groups; the signal is not completely continuous; instead there are two small spliced peaks. Furthermore, some subtle signals at wavenumbers between 2700–1750 cm$^{-1}$ which may correspond to tension $C-H$ vibrations. Vibrations at the 1650 cm$^{-1}$ wavenumber may correspond to the pattern of a mono-substituted amide (3500–3000 cm$^{-1}$, 1600 cm$^{-1}$, and 1400–1200 cm$^{-1}$) which also correspond to $C-N$ and $C=C$ bonds [21].

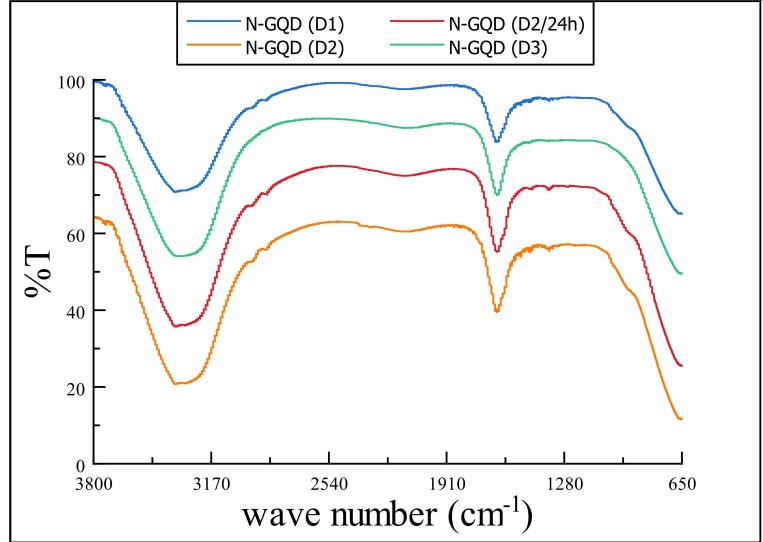

**Figure 2.** IR spectra for samples of N-GQD obtained using blank carbon (D1), MWCNT-OH (D2 and D2/24 h) and graphene (D3) after the hydrothermal treatment. Additionally, D2 Samples seems to be quite similar in every signal vibration (the y-axis values were modified to differentiate the spectra for each sample).

The UV-Vis absorption spectra for the products samples are very similar to each other; as we can see in Figure 3, the absorption bands are similar, such as the shoulder pike at

290 nm, followed by a broad peak at 340 nm. The signals below 300 nm are attributed to $\pi-\pi^*$ transitions of the aromatic domains; the broad one can be described as n–$\pi^*$ transition of multi-functional groups and $sp^2$ domains [15,16,18,21].

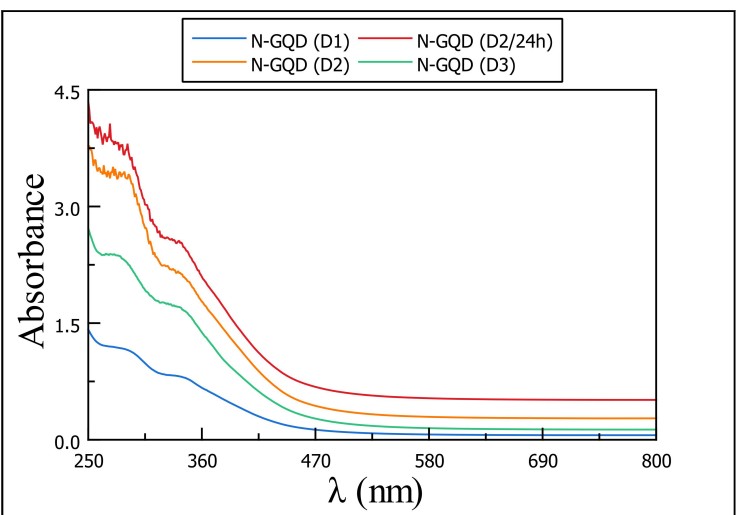

**Figure 3.** UV-Vis spectra for samples of N-GQD obtained using black carbon (D1), MWCNT-OH (D2 and D2/24 h) and graphene (D3) after hydrothermal treatment. Samples of D2 keep following very similar results (y-axis values were modified to differentiate the signals for each sample).

Microscopy analysis it is employed to observe the particle size and morphology. FE-SEM shows that, in general, small particles agglomerate to the precursor matter; this can be better seen in Sample D4 (Figure 4f) where clusters of graphene are clearly seen with small particles above them. It can be seen particles to 7 nm to 36.6 nm; apparently, Sample D1 (Figure 4a) shows better sizes particles, witch are the smallest ones with better distribution of size.

This analysis shows that a better process is needed to separate the small particles, but it could be easier to filter them because of the size difference between the desired product and the precursors.

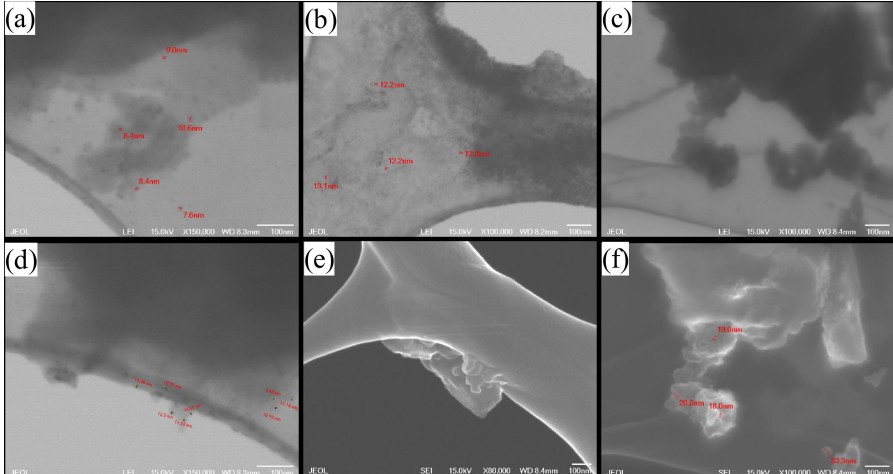

**Figure 4.** FE-SEM images before hydrothermal treatment for (**a**,**c**,**d**,**f**) 360 min and (**b**,**e**) 1440 min. Images (**f**,**e**) are generated by the backscattered electron modality. (**a**,**d**) images are for experiment D1, showing some agglomerated particles with size around of 8–10 nm; (**b**) for experiment D2, showing some agglomerated particles with size around of 11–13 nm; (**e**) for experiment D2/24 h; (**c**,**f**) for experiment D4, showing agglomerated particles with size around of 18–21 nm.

TEM analysis shows more accurate particle size information. The common observation for all samples is the way the hydrothermal treatment affects the precursor material; it's observable that in their layers, there are small circular particles with different linear patterns (0.21 nm bond distance, carbon distance) to those of the rest of the precursor material. The small particles sizes are between 4 and 7 nm. Black carbon material (Figure 5a,d) again shows better size distribution of small particles; MWCNT-OH (Figure 5b,c,e,f) in the analysis, it can be seen how tubes are opened because of the oxidative cleavage, forming a single layer of carbon or simply 'braking' the tube (Figure 5c). In addition, the results observable for experiment D1 are very similar to Lu et al. (2017) [16].

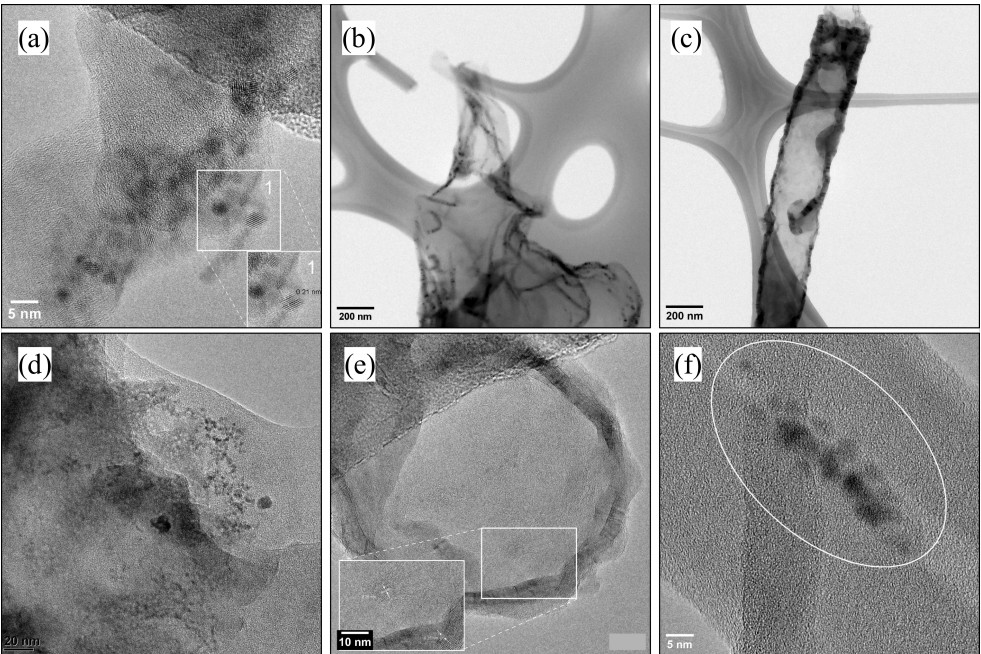

**Figure 5.** TEM images for Sample D1 (**a**,**d**), D2 (**b**,**e**) and D2/24 h (**c**,**f**). Analysis of the produced particle size and bond distance.

Figure 6 shows a very particular result: the particle size is much larger, and these, in turn, are agglomerated. According to the literature, the morphology of these structures is more similar to graphene flakes if guided by their size [22,23].

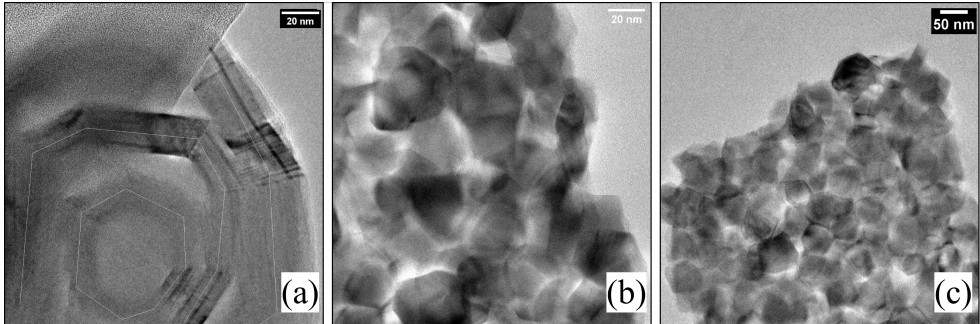

**Figure 6.** TEM images for experiment D3. White line in (**a**) follow an hexagonal structure, it is observable the small particle size agglomeration. For (**b**,**c**) images it is observable the characteristic hexagonal shape of graphene-like structures, these hexagonal particles with a size between 20–50 nm are agglomerated.

*3.2. Optical Properties of N-GQD*

The solution obtained after the hydrothermal procedure shown in Figure 7A had a color between brown and reddish. The solution exhibits greenish-blue fluorescence under a 365 nm UV lamp, pictured in the same figure.

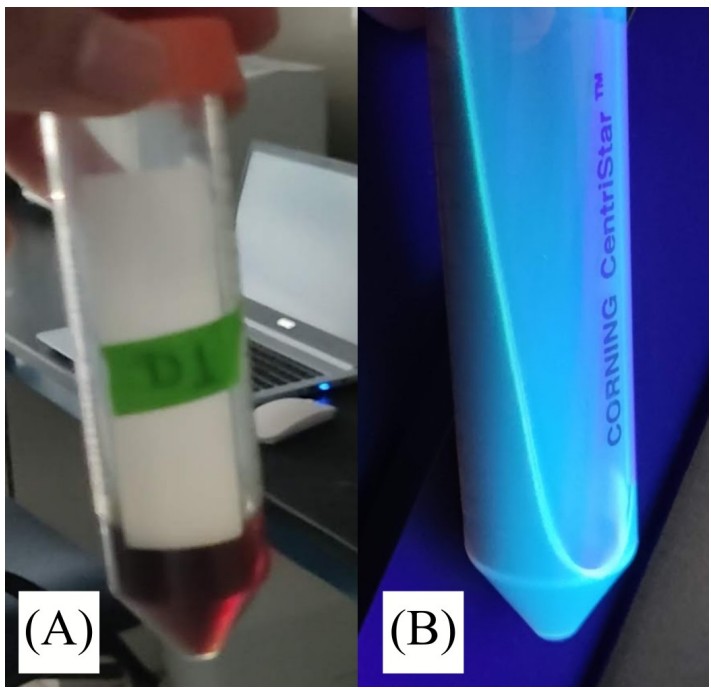

**Figure 7.** (**B**) figure show evidence of fluorescence in the sample obtained by hydrothermal treatment. The same solution color, figure (**A**), at daylight it's brown-reddish.

The results for photoluminiscense spectroscopy show a maximum emission peak at 496 nm, 490, nm and 488 nm for Samples D1, D2, D2/24 h, and D3 (the last two share the same maximum fluorescence wavelength) in Figure 8. Photoluminescence effect in N-GQDs is originated by the quantum confinement effect and the resonance of the double bonds; it can be affected by the particle size; and in the case of Sample D3, it's observable the shifting of the maximum emission peak in contrast to the rest of the samples when presenting different particle sizes (as mentioned in the TEM analysis).

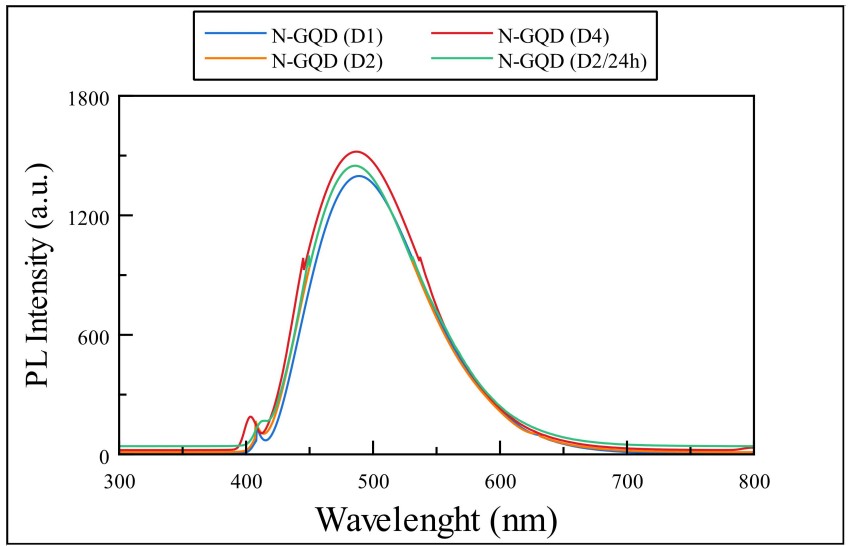

**Figure 8.** PL spectra samples obtained after hydrothermal treatment.

### 3.3. Chemical and Structural Characterization of N-GQD@AuNP

UV-Vis spectroscopy in Figure 9 shows a disappearance of the signal that was at the 250 nm wavelength, in addition to an increase in the signal at 300 nm and another small increase in the signal at 425 nm. The signal around 300 nm has been discussed as belonging to the N-GQD of the n-$\pi^*$ transitions of the C=O bonds. The second signal may belong to the possible coupling of gold due to its characteristic surface plasmonic resonance band, which is the reason for the color change. However, it is reported at 525 nm, where they demonstrate complete coupling. In the work of Ju et al. (2015) [11] and Chen et al. (2018) [5], they also report a shift to the left at 425 nm to report the coupling of the AuNP, so it is intuited that the fact of obtaining a yellow coloration and not a reddish color (as reported in most articles) means that there was no complete coupling [5,12]. We will wait to observe a better coupling to be able to characterize the composite in more detail.

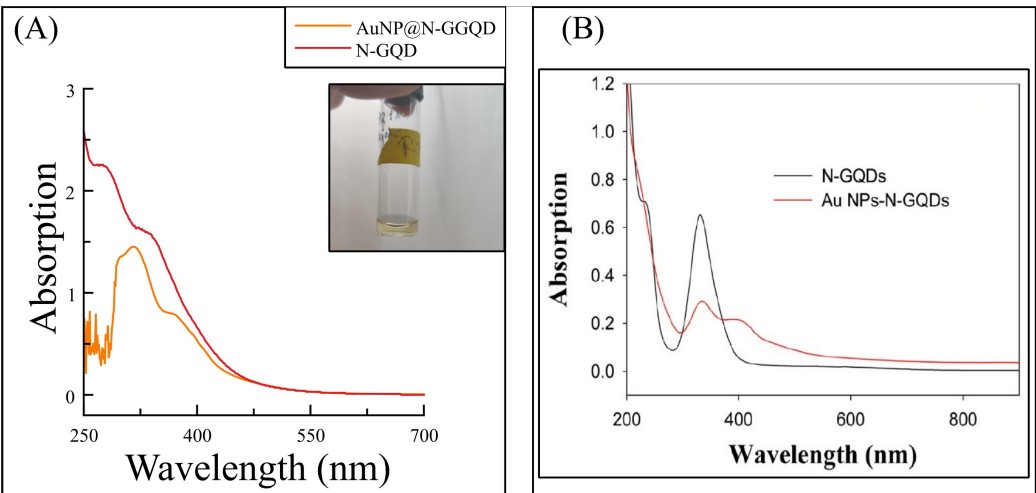

**Figure 9.** Result obtained from the coupling reaction of HAuCl$_4$ to N-GQDs (**A**) to form the AuNP@N-GQD nanocomposite; inserted: photograph of N-GQD@AuNP solution, contrasted against the work of Ju et al. (2015) [11] (**B**).

TEM images for the N-GQD@AuNP sample show in Figure 10 a distance bond of 0.24 nm, which probably corresponds to metallic AuNP with a particle size between 20 to 40 nm. In addition, it's observable that there's an agglomeration and different morphologies of the nanoparticles, indicating a possible partial coupling.

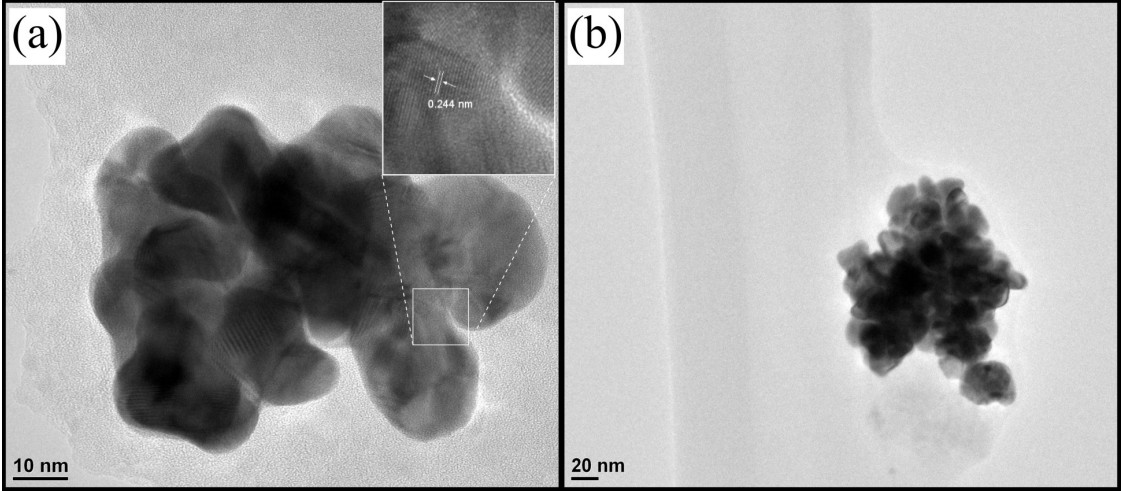

**Figure 10.** TEM images of the synthesized N-GQD@AuNP. (**a**) AuNP shown distance bond, and (**b**) it's observable the agglomeration with variable morphologies of the nanoparticles.

## 4. Conclusions

According to the XRD analysis, the obtaining of the nanometrial N-GQD with different carbon sources not reported in the literature is concluded. FT-IR analysis confirms that the synthesis of GQD with an N-modified chemical surface was carried out correctly, as it shows characteristic bands of amino and amide groups in the structure. The results in UV-Vis also demonstrates the presence of N groups in the characteristic bands at 290 nm and 340 nm that correspond to aromatic domains and ketones, and the blue shift (after the coupling of the AuNP) demonstrates that there was a contraction of nucleation, but it was not strong enough to show a more intense signal. Microscopy analysis demonstrate the particle size of N-GQD between 8 and 14 nm, the agglomeration of the particles, and the conclusion that it's necessary to add a purification process to the methodology. More analysis and experimentation time is required to explain some factor that has affected the nucleation of the AuNP to the GQD surface and to purify the reaction products. In summary, this methodology can be an alternative to obtaining N-GQD from accessible carbon sources and through a green reaction.

**Author Contributions:** The authors confirm contribution to the paper as follows: study conception and design: O.K. and I.G.; experimental procedure: D.I.; Analysis and interpretation of results: D.I., O.K. and I.G.; draft manuscript preparation: D.I. and I.G. All authors reviewed the results and approved the final version of the manuscript.

**Funding:** The authors acknowledge and express their gratitude to PAICYT and CONACYT for the finantial support by means of project 236CE-2022.

**Data Availability Statement:** All data derived from this research are discus and available in this document.

**Conflicts of Interest:** The authors declare no conflict of interest.

## Abbreviations

The following abbreviations are used in this manuscript:

| | |
|---|---|
| AuNP | Gold NanoParticles |
| GQD | Graphene Quantum Dots |
| N-GQD | Nitrogen-Graphene Quantum Dots |
| N-GQD@AuNP | Nitrogen-Graphene Quantum Dots coupled to Gold NanoParticles |
| FE-SEM | Field Emission-Scanning Electron Microscopy |
| TEM | Transmition Electron Microscopy |
| UV-Vis | Ultra-Violet spectroscopy |
| FT-IR | Fourier transform infrared |
| PL | photoluminisence |
| MWCNT-OH | Multi-Walled Carbon NanoTubes OH terminations |

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
