# Peer review of "Synthesis of Graphene Quantum Dots Coupled to Au Nanoparticles: A Facile and Versatile Route Using Different Carbon Sources"

_carbon, 2018_

Round 1
Reviewer 1 Report
In this submission, Ibarra et al, presents a top-down synthesis of graphene quantum dots (GQD) by hydrothermal method from different carbon sources (graphene, multi-walled carbon nanotubes and black carbon) and H2O2 as an oxidizing agent, in addition of N source added in the reaction to modify the chemical surface of the GQD, giving rise to the nanomaterial N-GQD. This is an interesting manuscript but a few mechanisms are not clear in its current format which I have listed below;
- H2O2 has extensively been reported in graphene research to introduce pores, so could please authors add the citations on this along with discussion that why H2O2 is needed and so on.
- Line 23 does not has a proper citation to support the statement, I recommend citing the paper; 10.2217/nnm-2018-0018
- Authors should add a table on comparison of this work to existing literature, since the novel synthesis of GQDs has extenisvly been reported. a table will help to understand how yield and other features are better in this work in comparison to previous works.
- TEM images have scale on different directions, please make it consistent.
Reviewer 2 Report
The authors prepared GQDs by top-down method which is uncommon in the literature and enriched the synthesis experience in this filed. Also, the dots showed strong fusion characteristics and be able to be coupled with gold nanoparticles. However, I suggest that this manuscript need to be revised before publication based on the following comments.
1. In the synthesis section, as mentioned by the authors in line 72, the weight ratio of carbon source and EDA is around 1 : 225. In this manuscript, I think EDA is used as a nitrogen source to surface functionalize GQDs. Then why such a large amount of EDA is needed for this reaction?
2. In this reaction condition, EDA could be oxidized by hydrogen peroxide to produce nitrogen containing organic species. To make the results more convincing, a control experiment under the same reaction conditions without carbon starting materials is suggested.
3. The manufactures and purity of all chemicals used in this manuscript is suggested to be included.
4. In figure 1, more detailed analysis may be helpful to understand the structure of carbon starting material. For example, why the x-ray diffraction position of (c) graphene is around 30 degree, but not around 26; why (a) black carbon contains diffraction peaks at 10 and 20 degree.
5. In figure 8, it will be appreciated if the authors could show the whole emission spectra. So, readers can have an overall understanding of the photoluminescence properties of GQDs made from three different carbon sources.
Reviewer 3 Report
Why was ethylenediamine chosen as the nitrogen source?
It is necessary to provide more information about the registration of the IR spectrum.
What determines the duration of hydrothermal synthesis, why exactly 6 hours?
Was complete removal of ethylenediamine achieved after synthesis, by evaporation on a rotary evaporator? If yes, how was it controlled?
Author Response
Please see the attachment. Thanks in advance.

Round 2
Reviewer 1 Report
Authors have elegantly addressed my comments and I am pleased to recommend this manuscript for publication.
Reviewer 2 Report
Thank you for the authors! My Concerns are solved.